# Dynamic Cooperation of Transportation and Power Distribution Networks via EV Fast Charging Stations

**Zihao Chen** [1,2] , **Bing Han** [1,*], **Fei Xue** [1], **Shaofeng Lu** [3] **and Lin Jiang** [2]

1. School of Advanced Technology, Xi'an Jiaotong-Liverpool University, Suzhou 215123, China
2. School of Electrical Engineering, Electronics and Computer Science, University of Liverpool, Liverpool L69 3GJ, UK
3. Shien-Ming Wu School of Intelligent Engineering, South China University of Technology, Guangzhou 510640, China
* Correspondence: bing.han@xjtlu.edu.cn

**Abstract:** With the development of electric vehicles, research on the cooperation of transportation networks (TNs) and power distribution networks (PDNs) has become important. Because of practicability, most cooperation research focuses on user equilibrium assignment based on the Wardrop I principle. There is less research focusing on network cooperation involving the system optimal assignment based on Wardrop II. This research paper constructs a cooperation between dynamic system optimal (DSO) and dynamic optimal power flow (DOPF) assignments with multi-objective optimization. Based on Wardrop II, this DSO model realizes multiple origin–destination pairs, multiple tasks, and multiple vehicle types. Electric vehicle and fast charging station models are designed as the connection between both networks. The optimal result gives three scenarios: TN prior, PDN prior, and a compromise of both. DSO minimized the total travel cost and DOPF minimized the total cost of power generation. Several path choices resulted from the scenarios. Whichever scenario is chosen, an electric vehicle is assigned dispersedly for a certain time period to reduce power loss. The optimal solution is also affected by the charging power in fast charging stations. This research can be applied to logistics transportation under traffic restrictions. It offers a dynamic optimization model for transportation and power operators.

**Keywords:** traffic assignment problem (TAP); dynamic traffic assignment (DTA); dynamic system optimal (DSO); optimal power flow (OPF); electric vehicle (EV); fast charging station (FCS)

## 1. Introduction

### 1.1. Background

The electric vehicle (EV) holds great promise for the coming decades. EVs allow for flexibility as gasoline vehicles (GVs), and at the same time, take measures against climate change by reducing the utilization of fossil fuel and the emissions of classical combustion engines [1]. By paving the way to a sustainable, low-carbon, and clean society, the EV is going to promote the integration of renewable energy and reduce the emission of carbon dioxide [2]. To reduce the dependence of fossil fuel, some countries have set clear targets for entering the generation of renewable energy and EVs and have promulgated market incentives to achieve these goals [3–6]. With the development and popularization of EVs and charging facilities, the interdependence between transportation networks (TNs) and power distribution networks (PDN) is currently boosting [7]. Power supply becomes a considerable matter in TNs that generate large power demand for vehicle operation. The development of EVs has greatly influenced TNs and will also impact PDNs. In the near-term outlook, there are almost 20 million passenger EVs on the road with 1.3 million commercial EVs covering buses, delivery vans, and trucks in 2022. The global sale of commercial EVs covering buses, delivery vans, and trucks more than doubled in 2021. In 2022, China had 685,000 electric buses on the road. By 2025, EVs will replace almost

2.5 barrels of oil per day and there will be 77 million passenger EVs on the road, which will be 6% of the total fleet. Some regions will implement these changes faster including China, estimated to reach 13%, and Europe, estimated to reach 8% [8]. The traditional traffic assignment problem (TAP), that is applied to the evaluation of GV's TNs, is not sufficient for the evaluation of EVs. Similar to the traditional GV, EVs also require energy supply on roads. On expressways with hundreds and thousands of links, the energy supply affects the path planning of EVs. Two aspects affecting the widespread use of EVs in transportation are charging infrastructure and battery technologies [9,10]. It is predicted that the battery demand of EVs will grow to 3486 GWh in the Economic Transition Scenario in 2030 [8]. The driving distance of EVs is still low because of the limit of battery capacity, which makes recharging an important player in the usage of EVs [11]. Expressways are equipped with charging stations for EVs. These fast charging stations (FCSs) are connected with PDNs. After 2030, stock shares of EVs are set to rapidly rise in all major markets in association with increased electrification of other energy services. Grid operators must have assurance that the necessary investments are executed to accommodate the increasing loads beyond 2030 and significant upgrades will probably be required [8]. To promote the management of EVs in addition to EV interactions with both TNs and the power system, the concept of coupled transportation power systems is proposed [12]. The rapid proliferation of EVs and their charging in FCSs could affect TNs, PDNs, as well as the interactions [13]. Unreasonable path planning will cause traffic congestion and increase users' travel costs. Congestion patterns and regulation policies in TNs affect the driving patterns of EVs. The role of the EV will influence the distribution of traffic flow in TNs, and the congestion caused by an increase in EVs will result in a large demand for charging stations to PDNs, potentially causing an overload in power demands for buses and lines. The charging time of EVs will also influence the traffic distribution of EVs and even TNs [12,14–19]. The EV state of charge (SOC), charging service fee, and the number of EV charging piles impact TNs significantly [20]. In the future, with the rapid popularization of EVs, this phenomenon is going to be more common. Operators of power systems should be able to balance supply and demand of the power grid at any time, which requires sufficient power resources through the generation or storage of, ample network capacity [21]. Therefore, it is necessary to consider cooperation between the TN and the PDN.

It is not necessarily straightforward to upgrade the PDN to accommodate the needs of an increased number of EVs [21]. The charging activities of EVs can be affected by the congestion of power transmission lines, which will change the distribution of traffic flow in the TN [7]. For instance, a PDN could possibly deliver high levels of power demand but some lines may be limited because of small capacity due to low loads and low predicted load growth, such as in rural areas compared with urban areas. Clustering effects could cause problems even at low EV uptake, particularly in residential regions [21]. The upgrading of a power grid to fit the TN is also a cooperation problem for electricity operators.

### 1.2. Literature Review

The recent Traffic Assignment Problem (TAP) is based on two principles: Wardrop I and Wardrop II [22]. The former is a user equilibrium (UE) describing an equilibrium state in which any single traveler cannot change their state on their own. The latter principle is a System Optimal (SO) strategy for the transportation manager. Wardrop I describes that while all drivers know precisely the traffic situation of a TN and try to choose the shortest path, the TN will reach a state of equilibrium in which any single driver cannot change their state by changing their path or travel behaviors. This is usually a spontaneous user decision. Wardrop II considers another situation with participation of the transportation manager to minimize the total travel cost in a transportation system. The respective dynamic patterns of UE and SO correspond to dynamic user optimal (DUO) and dynamic system optimal (DSO). Beckmann provided the first mathematical optimization model solving UE [23]. UE and SO are both static TAPs.

The dynamic traffic assignment (DTA) model developed by Merchant and Nemhauser raised a mathematical programming method to deal with DSO [24,25]. This is the first DTA model based on mathematical programming and it is called the M–N model. The model is discrete, nonconvex, and nonlinear [26]. Ho gave a continuous linear optimization method [27]. Craey reformed the M–N model into a nonlinear convex programming model [28]. Ran and Boyce developed an optimal control model for DUO patterns that is the dynamic model of UE [29]. Table 1 illustrates the relationship of the four concepts.

**Table 1.** Class of TAP.

|                | Wardrop I | Wardrop II |
|----------------|-----------|------------|
| TA (static)    | UE        | SO         |
| DTA (dynamic)  | DUO       | DSO        |

There are some studies about the combination of TNs and PDNs. A mix-integer quadratically constrained programming (MIQCP) method to realize cooperation between the two networks is provided in [13]. A generalized UE method for coupled power-TN operation is proposed in [19]. A nonlinear complementarity programming model cooperating UE with OPF considering Locational Marginal Price (LMP), path selections, charging opportunities, and individual rationality of minimum travel cost in a convex TAP over an extended TN is given in [7].

Most of the works previously mentioned are static and focus on cooperation between the UE and the power system. Newell thought the static TAP method could easily generate mistakes and Ben-Akiva thought the static TAP model was not effective at analysis of traffic congestion [30,31]. The DTA model considers the variation in departing and arriving travel times, nonuniformity of traffic flow, variation of shortest path to an Origin–Destination (O–D) pair, and variation of congestion time and location [26]. In a realistic traffic system, vehicles travel dynamically through the TN and are mutually dependent on the path choices [1]. A stochastic, bi-level, and simulation-based decision-making framework for prioritizing mitigation and repair resources to maximize the expected resilience improvement of an interdependent DUO-electric power system under budgetary constraints is presented in [32]. A strategy to deal with semi-dynamic transportation problems is provided in [33]. A multi-objective programming model solved with a new bilayer Benders decomposition algorithm is constructed in [34]. A dynamic interaction between DUO and OPF is realized by updating the number of vehicles and LMP through optimization iteration in [35]. An integrated modeling framework for the real-time operation and analysis of system cooperation is developed in [36]. A stochastic multi-agent simulation-based model with the objective of minimizing the total cost of interdependent TNs and PDNs is solved in [37]. A DTA addressing the operation of EVs including their range limitations caused by limited battery energy and necessary recharging stops is studied in [1]. Especially, a DTA model is exploited to account for the time-varying travel demand and flow dynamics. A novel optimal traffic power flow problem to analyze the spatial and temporal congestion propagation on coupled systems—under congested roads, transmission lines, and FCSs is proposed in [7].

These studies consider dynamic TN–PDN cooperation within the Wardrop I principle, including the MIQCP model, bilevel stochastic model, multi-objective programming model, updating and iteration method, and multi-agent simulation-based model. The operation methods include optimization programming and simulation. Some papers are the dynamic extension of static models. For example, ref. [35] adopts the solution updating method [12] and uses DUO to substitute the original UE model. Wardrop I is commonly used to describe users' spontaneous behaviors. However, it cannot be applied to scenarios with the participation of a transportation manager, such as logistic transportation; Wardrop II can be applied to such scenarios.

Some studies do not involve TAP; however, EVs provide an effect to the PDN in the area of power grid resilience and cyber attacks. To realize economic effectiveness in load frequency control (LFC) while sustaining satisfactory system performance, a distributed economic model predictive control strategy is proposed in [38] for the LFC considering large-scale plug-in electric vehicles (PEVs). For the characteristics of fast charging and discharging of PEVs, a coordinated approach to PEVs in a conventional LFC is given in [39]. To reduce communication pressure and solve the problems of cyber attacks effectively, a resilient event-triggered mechanism is adopted in individual EV-charging price updates and transmission in [40]. An optimization model for joint post-disaster PDN restoration considering coordinated electric bus dispatching is proposed in [41].

*1.3. Main Contributions*

The scientific aim of this research paper is to construct a TN and PDN dynamic cooperation model referring to DSO and DOPF. This article brings two main novelties to the existing literature in the following areas:

- Compared with the former literature review, it constructs a DSO model based on the Wardrop II principle. This model involves transportation and electricity operators and realized cooperation between both operators. Different from the spontaneity of vehicle drivers described in Wardrop I, Wardrop II provides an opportunity for the transportation operator to dispatch vehicles at various time intervals.
- In addition, compared with the common DTA model, the model not only considers multiple O–D pairs and multiple vehicle types, but also multiple tasks. Multi-task is a middle class under multiple O–D pairs and over vehicle types. This class determines various departing times. It is useful in the logistical department. It can assign the departing and arriving times of transportation for various goods.

Finally, this article is organized as follows: Section 2 provides the mathematical model. It includes the TN model, PDN model, FCS model, and multi-objective model. Section 3 provides the results of a case study. Section 4 is the conclusion.

## 2. Mathematical Model

*2.1. Model Structure*

This research realizes a cooperation between DSO and dynamic optimal power flow (DOPF) through multi-objective optimization. It consists of three sub-models:

- DSO (modeling of TN)
- DOPF (modeling of PDN)
- FCS (modeling connecting two networks)

TN and PDN are connected by FCS as shown in Figure 1. The FCS is designed as the connection of two networks. The power demand in the FCS connects the vehicle flow and the power flow. The vehicle flow into the FCS causes a variation in power demand which influences the power flow in the PDN.

The DSO model proposed in this paper is based on the original TN model in the chapter 4 of [29]. Based on the original TN model, the new model adds task and vehicle type as two factors to construct multiple O–D pairs, multiple tasks, multiple vehicle types and paths into four classes of vehicle assignment in the TN model. This approach guarantees strong first-in-first-out (SFIFO) constraints. A DSO objective function is added to the model. The DOPF model is modified from the common Quadratically Constrained Programming (QCP) OPF model with voltage phase angle relaxation. The model of the FCS is designed based on the linear relationship between the power demand and the charging vehicle number in [12]. It considers the accumulation of the number of EVs and the charging time is related to the vehicle type and the power of the charging pile.

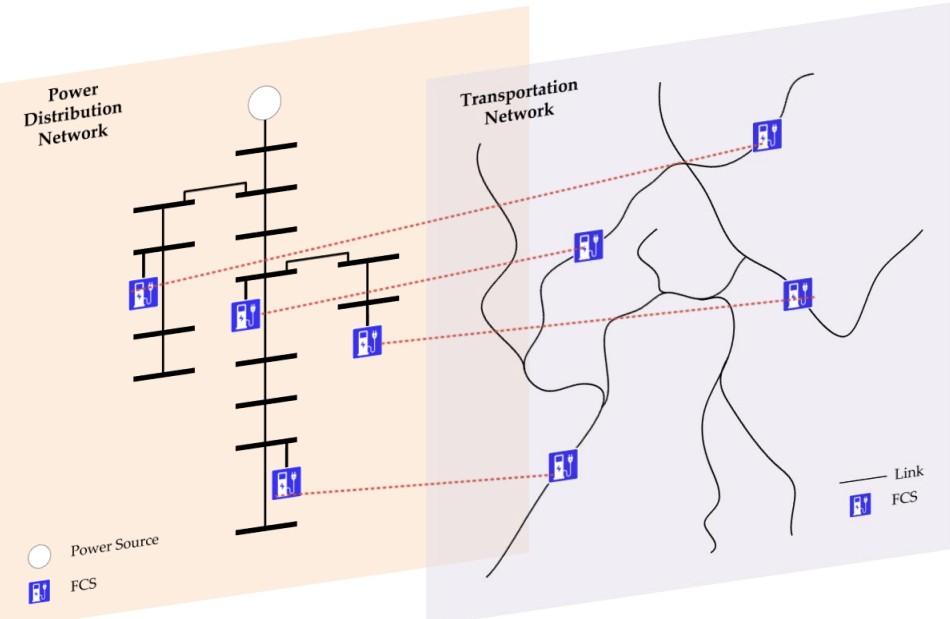

**Figure 1.** Conceptual of the coordination of TN and PDN.

The DSO is based on the Wardrop II principle. It minimizes the total travel cost of the TN by selecting paths for travelers based on time period 0 to T. It is devised for the transportation manager and utilized for logistics transportation with traffic restriction. For example, some locations may enable power priority at certain times (in the case of events such as war or other emergencies). In such cases, the transportation manager needs to limit some links in order to form a restriction in the TN. With the addition of EVs, the power supply of EVs must be ensured by the power system operator. This project can help both transportation and power system departments to optimize vehicle departing and arriving and power flow dispatch.

### 2.2. Modeling of TN

DSO is a linear programming (LP) model. The DSO explored in this paper considers path factor and SFIFO. For a TAP, this model considers various O–D pairs $rs$, various tasks $h$, and various vehicle types $m$, including GVs and EVs. It also considers the battery capacity of EVs $E_m$ and the speed of various vehicles $sp_m$. Path $q$ includes various links on a TN and at a FCS $i$ ($i$ is the bus connecting to a PDN). For GVs, the battery capacity $E_m$ is 0 and the paths $q$ do not pass FCSs. Even though GVs and EVs have the same link that has a FCS, the paths $q$ is different. Four classes of vehicle assignment logic are designed. The top class of vehicle assignment is O–D pair $rs$. The second class is various task $h$. For the same O–D pair $rs$, various tasks can be assigned to different time periods. The third class is vehicle type $m$. A task can be carried out by various types of vehicles. Path $q$ is the lowest assignment class. In the same task, various vehicles are assigned to different paths and time intervals.

Equation (1) is the objective function. $t$ is the index of each time interval and $x_a(t)$ is the number of vehicles on a link $a$ at time $t$. $q$ is the sum of all paths passing through a link $a$ at time interval $t$. Its object is to minimize the total travel cost of TN. It is the sum of the travel cost of in the TN for a time period 0 to T. While a vehicle is on a link $a$, the vehicle has a cost $c_a$ to travel from the entrance to the exit of the link. For a more realistic explanation, the travel cost can be the cost of time, fuel, or electric energy.

$$\min_{x_a(t)} \sum_{t=0}^{T} \sum_{a} c_a[x_a(t)] \tag{1}$$

Figure 2 illustrates the state and control variables on link $a$. For vehicle type $m$ in task $h$ over path $q$ passing through link $a$ with O–D pair $rs$ at time interval $t$, a link $a$ has a state of variables: the number of vehicles $x^{rs}_{ahmq}(t)$ and two control variables, the inflow rate $u^{rs}_{ahmq}(t)$ and the exit flow rate $v^{rs}_{ahmq}(t)$. The inflow rate $u^{rs}_{ahmq}(t)$ is the number of vehicles entering the link at time $t$ and the exit flow rate $u^{rs}_{ahmq}(t)$ is the number of vehicles exiting the link at time $t$.

$$u^{rs}_{ahmq}(t) \quad \overline{\quad\quad x^{rs}_{ahmq}(t) \quad\quad} \quad v^{rs}_{ahmq}(t)$$

**Figure 2.** Flow variables for link $a$.

Equation (2) is the state equation for link $a$. For vehicle type $m$ in task $h$ over path $q$ passing through link $a$ with O–D pair $rs$ at time interval $t$, the number of vehicles $x^{rs}_{ahmq}(t+1)$ at next time $t+1$ depends on the number of vehicles $x^{rs}_{ahmq}(t)$, the inflow rate $u^{rs}_{ahmq}(t)$, and the exit flow rate $v^{rs}_{ahmq}(t)$ at this time.

$$x^{rs}_{ahmq}(t+1) = u^{rs}_{ahmq}(t) + x^{rs}_{ahmq}(t) - v^{rs}_{ahmq}(t) \tag{2}$$

Equation (3) assigns the instantaneous departing flow $f^{rs}_{rhm}(t)$ at the origin $r$ to vehicle $m$ in task $h$. The total number of vehicles $m$ departing from origin $r$ of the O–D pair $rs$ $F^{rs}_{rhm}$ is equal to the total assigned vehicle number $F^{rs}_{hm}$. $F^{rs}_{rhm}$ is assigned to different times $f^{rs}_{rhm}(t)$. Instantaneous departing flow $f^{rs}_{rhm}(t)$ is the number of vehicles $m$ departing from origin $r$ of O–D pair $rs$ at time $t$. The final time T is not included in the assignment because all the vehicles should depart from the origin $r$ before that time.

$$F^{rs}_{rhm} = \sum_{t=0}^{T-1} f^{rs}_{rhm}(t) \tag{3}$$

Equation (4) describes the instantaneous arriving flow $f^{rs}_{shm}(t)$ at the destination $s$ to vehicle $m$ in task $h$. The total number of vehicles $m$ in task $h$ arriving for time period 1 to T $F^{rs}_{shm}$ is equal to the total assigned vehicle number $F^{rs}_{hm}$. The instantaneous arriving flow $f^{rs}_{shm}(t)$ is the number of vehicles $m$ arriving at destination $s$ of O–D pair $rs$ at time $t$. $F^{rs}_{shm}$ is the sum of the instantaneous arriving flow $f^{rs}_{shm}(t)$ for time period 1 to T. At the initial time $t = 0$, $f^{rs}_{shm}(t) = 0$ because none of the vehicles can move from origin $r$ to destination $s$ instantaneously.

$$F^{rs}_{shm} = \sum_{t=1}^{T} f^{rs}_{shm}(t) \tag{4}$$

Equation (5) and Figure 3a illustrate the connection that $f^{rs}_{rhm}(t)$ is assigned to the inflow rate $u^{rs}_{ahmq}(t)$ of different paths $q$. At any interval $t$, the instantaneous departing flow $f^{rs}_{shm}(t)$ at origin $r$ is assigned to different paths $q$.

$$f^{rs}_{rhm}(t) = \sum_{a \in B_r} u^{rs}_{ahmq}(t) \tag{5}$$

In Figure 3a, the dashed line denotes the link connecting to origin $r$ but does not belong to the path set of vehicle $m$ in task $h$ with O–D pair $rs$. The vehicles are assigned only to the links denoted by the full lines. $\forall\, a \notin q$ is the dashed denoted link, $u^{rs}_{ahmq}(t) = 0$, $x^{rs}_{ahmq}(t) = 0$, and $v^{rs}_{ahmq}(t) = 0$. In another word, the dash line denoted links are unavailable to some assignments of specific vehicles $m$ in task $h$ with O–D pair $rs$. However, these links are connected to the node and it is possible for them to be available to assignments of other vehicle types, other tasks or other O–D pairs.

Equation (6) and Figure 3b give the FCC at destination $s$. At any interval $t$, the instantaneous arriving flow $f_{shm}^{rs}(t)$ at destination $s$ is the sum of the exit flow rate $v_{ahmq}^{rs}(t)$ of different paths $q$. $\forall\, a \notin q$ is the dashed denoted link, $u_{ahmq}^{rs}(t) = 0$, $x_{ahmq}^{rs}(t) = 0$, and $v_{ahmq}^{rs}(t) = 0$.

$$f_{shm}^{rs}(t) = \sum_{a \in A_s} v_{ahmq}^{rs}(t) \tag{6}$$

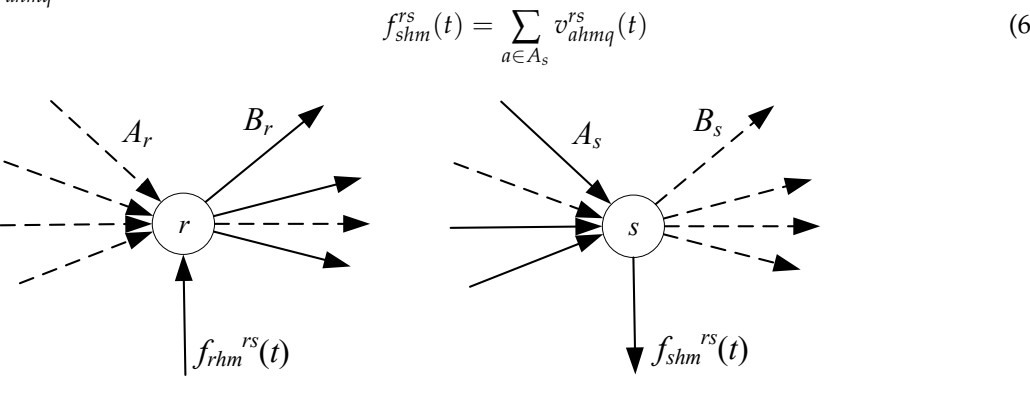

(**a**)　　　　　　　　　　　　　　　　(**b**)

**Figure 3.** (**a**) Flow conservation constraints (FCC) at origin $r$ of vehicle $m$ in task $h$; (**b**) FCC at destination $s$ of vehicle $m$ at task $h$.

Equations (3)–(6) give the allocation of traffic flow to time intervals and paths. An allocation unit is determined by the O–D pair $rs$, task $h$, and vehicle type $m$. Path $q$ is at the lowest class of allocation. The allocation method is: the total vehicle number $F_{hm}^{rs}$ in a task is assigned to instantaneous flow at various time intervals, $f_{rhm}^{rs}(t)$ and $f_{shm}^{rs}(t)$, and then instantaneous flow is assigned to different paths $u_{ahmq}^{rs}(t)$ and $v_{ahmq}^{rs}(t)$. The allocation process can be expressed as below.

$$F_{rhm}^{rs} \rightarrow;\ f_{rhm}^{rs}(t) \rightarrow u_{ahmq}^{rs}(t) \dots v_{ahmq}^{rs}(t) \rightarrow f_{shm}^{rs}(t) \rightarrow F_{shm}^{rs}$$

Equation (7) describes other nodes $l$ besides the O–D pair $rs$. At a common node $l$ over a path $q$, there is not an instantaneous departing flow $f_{shm}^{rs}(t)$ or an instantaneous arriving flow $f_{shm}^{rs}(t)$. The sum of the exit flow rate $\sum_{a \in A_l} v_{ahmq}^{rs}(t)$ is equal to the sum of the $\sum_{a \in B_l} u_{ahmq}^{rs}(t)$ inflow rate at any time $t$.

$$\sum_{a \in A_l} v_{ahmq}^{rs}(t) = \sum_{a \in B_l} u_{ahmq}^{rs}(t), \quad \forall l \neq r \cap l \neq s \tag{7}$$

Figure 4 illustrates Equation (7). Even though the double sides of (7) are a sum operation, there is only one link for inflow and one link for exit flow over one path $q$ because it is not necessary for a path to pass through a node twice or more.

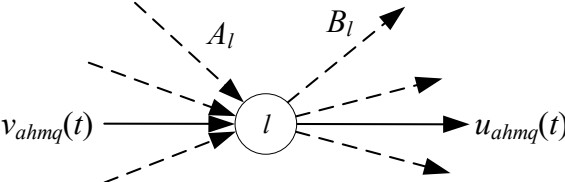

**Figure 4.** FCC at common node $l$ excluding origin $r$ and destination $s$ of vehicle $m$.

$\forall\, a \notin q$ is the dashed denoted link, $v_{ahmq}^{rs}(t) = 0$ and $u_{ahmq}^{rs}(t) = 0$.

Equation (8) shows the Flow Propagation Constraint (FPC) of this model. This FPC guarantees SFIFO constraints in which for the same type of vehicle, the flow entering later cannot catch up or even overtake the flow entering earlier. If there are $u_{ahmq}^{rs}(t)$ vehicles

$m$ in task $h$ inflow link $a$ over path $q$ of O–D pair $rs$ at time $t$, considering the distance $d_a$ and the driving speed $sp_m$, the link travel time (neglecting congestion) $\tau_{am}$ given by Equations (9) and (10), this group of vehicles $m$ would exit at time $t + \tau_{am}$.

$$u_{ahmq}^{rs}(t) = v_{ahmq}^{rs}(t + \tau_{am}) \tag{8}$$

For path $q$ not passing through the charging station on link $a$

$$\tau_{am} = \frac{d_a}{sp_m} \tag{9}$$

For path $q$ passing through the charging station on link $a$

$$\tau_{am} = \frac{d_a}{sp_m} + \mathrm{T}_{am}^{ch} \tag{10}$$

$\mathrm{T}_{am}^{ch}$ is the charging time of EVs $m$ in the FCS on link $a$. It is equal to the battery capacity $E_m$ over the real charging power of the charging pile in the FCS.

$$\mathrm{T}_{am}^{ch} = \frac{E_m}{p_i^{ch}} \tag{11}$$

In order for EVs with a long path length to complete travel without charging, the path must include a link with a FCS, even considering the time required for charging. For long distance travel or EVs with high-power consumption, the number of charging times would be more than one.

Equations (12)–(14) provide the total inflow rate $u_a(t)$, the total number of vehicles $x_a(t)$, and the total exit flow rate $v_a(t)$ on link $a$. These are the sum of path $q$, vehicle type $m$ at task $h$, and O–D pair $rs$. An O–D pair $rs$ includes various vehicle types $m$. The path is designed to refer to vehicle type.

$$u_a(t) = \sum_{rs} \sum_h \sum_m \sum_q u_{ahmq}^{rs}(t) \tag{12}$$

$$x_a(t) = \sum_{rs} \sum_h \sum_m \sum_q x_{ahmq}^{rs}(t) \tag{13}$$

$$v_a(t) = \sum_{rs} \sum_h \sum_m \sum_q v_{ahmq}^{rs}(t) \tag{14}$$

Equation (15) is a constraint. It limits that over a path $q$, the exit flow rate cannot exceed the vehicle number in the link $a$. Otherwise, it would be unreasonable.

$$x_{ahmq}^{rs}(t) \geq v_{ahmq}^{rs}(t) \tag{15}$$

### 2.3. Modeling of PDN

The model of PDN is realized by DOPF. This model includes a quadratic constraint. Compared with the standard OPF model, it refers to voltage phase angel relaxation. Equation (16) is the objective function of this model.

$$\min_{P_g(t)} \sum_{t=0}^{\mathrm{T}} \sum_{g=1}^{G} c_g[P_g(t)] \tag{16}$$

$c_g[P_g(t)]$ is the generating cost function versus the real power generation $P_g(t)$ of the generating unit $g$ at time interval $t$. Commonly, its form includes linearity, quadratic, cube, and piecewise. Quadratic is common in transmission systems and linear is common in distribution systems.

Equations (17) and (18) decouple complex power flow of branch $ij$ into real power flow $P_{ij}(t)$ and reactive power flow $Q_{ij}(t)$. If bus $i$ is the input end and bus $j$ is the output end of branch $ij$, (17) describes the input end (from bus) and (18) describes the output end (to bus). For an end, taking bus $i$ as an example, the sum of the square of the real power flow $P_{ij}^2(t)$ and the square of the reactive power flow $Q_{ij}^2(t)$ is equal to the product of the square of the voltage magnitude $V_i^2(t)$ at this end and the square of the current magnitude $I_{ij}^2(t)$. $V_i^2(t)$ and $I_{ij}^2(t)$ are taken as independent variables in this model. The ends of power flow input end and output end are self-defined. Both ends can be defined as the real input end or the output end. If one end is defined as the input but power flow output is actually from that end, its power flow in mathematics is negative, otherwise, it is positive. There is not sign constraint in the real and reactive power flow of both ends in this model.

$$P_{ij}^2(t) + Q_{ij}^2(t) = V_i^2(t)I_{ij}^2(t), \quad \forall i \in fb \tag{17}$$

$$P_{ji}^2(t) + Q_{ji}^2(t) = V_j^2(t)I_{ij}^2(t), \quad \forall i \in tb \tag{18}$$

Equation (19) provides the capacity limit of the transmission line. $S_{ijmax}$ is the complex power flow limit of branch $ij$.

$$P_{ij}^2(t) + Q_{ij}^2(t) \leq S_{ijmax}^2, \quad \forall i \in fb \cup tb \tag{19}$$

Equation (20) provides the power balance equation of real power and reactive power. To bus $j$, the net of the total real power input is equal to the total real power generated minus the constant load $P_j^d$ and the load demand of the FCS connecting to the bus $P_j^{ch}(t)$. Here $P_j^g(t) \geq 0$ and $P_j^{ch}(t) \geq 0$.

$$\sum_{j \in tb} P_{ij}(t) + \sum_{j \in fb} P_{jk}(t) = \sum_{g \in j} P_j^g(t) - P_j^d - P_j^{ch}(t) \tag{20}$$

Equation (21) provides the power balance equation of real power and reactive power. Compared with (20), there is a reactive compensation item $V_j^2(t)b_j$ in the formula. $b_j$ is the total shunt capacity connecting bus $j$. It is half of the sum of the shunt capacity of branches connecting to the bus. The other difference is $Q_j^g(t)$ and $Q_j^{ch}(t)$ can be less than zero. For $b_j > 0$ and $V_j^2(t) > 0$, $V_j^2(t)b_j > 0$. Reactive compensation in a system decreases reactive power output in a system.

$$\sum_{j \in tb} Q_{ij}(t) + \sum_{j \in fb} Q_{jk}(t) = \sum_{g \in j} Q_j^g(t) + V_j^2(t)b_j - Q_j^d - Q_j^{ch}(t) \tag{21}$$

Equations (22) and (23) provide the calculation of the real and reactive power loss in branch $ij$. $r_{ij}$ is the resistance of branch $ij$. $x_{ij}$ is the reactance of branch $ij$. The power loss is to add the power flow of both sides $i$ and $j$ together. In a specific branch and a specific time interval, the real or reactive power flow is certain from one side to the other side. If one side's real or reactive power flow is positive (negative), the other side must be negative (positive) and the absolute real or reactive power flow in the positive side must be larger than the negative side for the positive side is power input and the negative side is power output. The branch losses some power from the power input. Because of the opposite of both sides, the real or reactive power loss is the sum of both sides.

$$P_{ji}(t) + P_{ij}(t) = I_{ij}^2(t)r_{ij} \tag{22}$$

$$Q_{ji}(t) + Q_{ij}(t) = I_{ij}^2(t)x_{ij} \tag{23}$$

Equation (24) gives the square difference of voltage magnitude of both sides of a branch. This equation includes voltage phase angle relaxation, which turns the traditional

complex OPF model into one that includes mere quadratic constraints and linear constraints, decreasing the difficulty of the solver.

$$V_i^2(t) - V_j^2(t) = 2\left[r_{ij}P_{ij}(t) + x_{ij}Q_{ij}(t)\right] - I_{ij}^2(t)\left(r_{ij}^2 + x_{ij}^2\right) \tag{24}$$

If there is no capacity limit in the power transmission line, it does not have any other influence on the optimal result of the DOPF. However, most power system cases ignore the capacity limit. It is not usual that the power flow exceeds the capacity limit of the branch or transformer. The standard OPF model of the phase angle includes complex constraints and not standard. It is a mix type, including triangular constraints and high order items. Converting it to a linear + quadratic constraints model is easy for optimization.

The dynamic OPF model should have the constraints of ramps $P_g^{ramp}$ and $Q_g^{ramp}$, as shown in Equations (25) and (26). The power generation change of a generating unit $g$ cannot exceed the limit in one time interval.

$$\left|P_g(t+1) - P_g(t)\right| \le P_g^{ramp} \tag{25}$$

$$\left|Q_g(t+1) - Q_g(t)\right| \le Q_g^{ramp} \tag{26}$$

*2.4. Modeling of FCS*

The following section addresses FCSs. There are two constraints connecting the DSO and DOPF models.

Equations (27) and (28) give the load demand versus the number of EVs in the FCS. $P_i^{ch}(t)$ is the real power demand of the FCS connecting to bus $i$ at time $t$. $p_i^{ch}$ is the real charging power of a charging pile at the FCS connecting to bus $i$.

$$P_i^{ch}(t) = p_i^{ch} \sum_{a \in i} x_a^{ch}(t) \tag{27}$$

$$x_a^{ch}(t) = \sum_{rs}\sum_{h}\sum_{m}\sum_{q \in i}\sum_{t_{am}^{ch}=0}^{T_{am}^{ch}-1} v_{amhq}^{rs}\left(t + t_{am}^{ch} + \tau_{am}^{\gamma}\right) \tag{28}$$

$x_a^{ch}(t)$ is the vehicle number on link $a$ charging at the FCS at time $t$. $T_{am}^{ch}$ is the total charging time of the EV $m$ at the FCS on link $a$. $t_{am}^{ch}$ is the charging time interval index of vehicle $m$ at the FCS on link $a$. $t_{am}^{ch} = 0$ represents those EVs of type $m$ which have finished charging and will leave the FCS at this time $t$ immediately. $t_{am}^{ch} = T_{am}^{ch} - 1$ represents those EVs of type $m$ which enter the FCS at time $t - 1$ and will spend $T_{am}^{ch}$ to charge since time $t$. $\sum_{t_{am}^{ch}=0}^{T_{am}^{ch}-1}$ represents adding all EVs $m$ on link $a$ charging at the FCS together according to the remaining charging time $t_{am}^{ch}$. $\tau_{am}^{\gamma}$ is the link travel time of vehicle $m$ from the FCS on link $a$ to the link's exit without congestion. It is about the distance from the FCS down link $a$ to the link's exit and the speed of the EV $m$. Equation (29) provides an introduction of $\tau_{am}^{\gamma}$. $\sum_{q \in i}$ is used to judge if the path $q$ in the task $h$ passes through the FCS down link $a$.

$$\tau_{am}^{\gamma} = \frac{d_a^{\gamma}}{sp_m} \tag{29}$$

Equation (30) is the number of EVs limited to one side of the FCS. $x_{amax}^{ch}$ is the number of charging piles of one side of the FCS.

$$x_a^{ch}(t) \le x_{amax}^{ch} \tag{30}$$

Figure 5 provides an example of FCS. There is a pair of opposite links between node 1 and node 2. One FCS is located at the two links. Both sides of links belong to the FCS that is connecting to the same bus in the PDN.

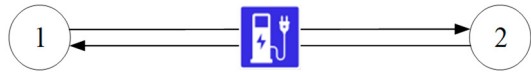

**Figure 5.** Representation of a FCS.

Table 2 and Figure 5 provide data for an example travel scenario. Suppose there is a task from node 1 to node 2. Six of the same type of EVs are travelling and all require charging at the FCS. The travel time from node 1 to the FCS and the time from the FCS to node 2 are equal to 2 h. The charging time of the 6 EVs is 3. For time period 0 to 7, the 6 EVs can finish the travel period. $\tau_{am}^{\gamma} = 1$ and $T_{am}^{ch} = 3$. The 6 vehicles are assigned to times 0, 1, and 2 corresponding to 1, 2, and 3 vehicles to depart.

**Table 2.** Example of FCS vehicle number statistics.

| | $f_{1hm}^{12}(0)$ | $f_{1hm}^{12}(1)$ | $f_{1hm}^{12}(2)$ | $f_{1hm}^{12}(3)$ | $f_{1hm}^{12}(4)$ | $f_{1hm}^{12}(5)$ | $f_{1hm}^{12}(6)$ | $f_{1hm}^{12}(7)$ |
|---|---|---|---|---|---|---|---|---|
| | 1 | 2 | 3 | | | | | |
| | $v_{ahmq}^{rs}(0)$ | $v_{ahmq}^{rs}(1)$ | $v_{ahmq}^{rs}(2)$ | $v_{ahmq}^{rs}(3)$ | $v_{ahmq}^{rs}(4)$ | $v_{ahmq}^{rs}(5)$ | $v_{ahmq}^{rs}(6)$ | $v_{ahmq}^{rs}(7)$ |
| $x_i^{ch}(0) = 0$ | | 0 | 0 | 0 | | | | |
| $x_i^{ch}(1) = 0$ | | | 0 | 0 | 0 | | | |
| $x_i^{ch}(2) = 1$ | | | | 0 | 0 | 1 | | |
| $x_i^{ch}(3) = 3$ | | | | | 0 | 1 | 2 | |
| $x_i^{ch}(4) = 6$ | | | | | | 1 | 2 | 3 |
| $x_i^{ch}(5) = 5$ | | | | | | | 2 | 3 |
| $x_i^{ch}(6) = 3$ | | | | | | | | 3 |
| $x_i^{ch}(7) = 0$ | | | | | | | | |
| | $f_{2hm}^{12}(0)$ | $f_{2hm}^{12}(1)$ | $f_{2hm}^{12}(2)$ | $f_{2hm}^{12}(3)$ | $f_{2hm}^{12}(4)$ | $f_{2hm}^{12}(5)$ | $f_{2hm}^{12}(6)$ | $f_{2hm}^{12}(7)$ |
| | | | | | | 1 | 2 | 3 |

### 2.5. Multi-Objective Optimization

Equations (1)–(15) are the TN model. Equations (16)–(26) are the PDN model and Equations (27)–(30) are the FCS model.

Finally, this problem is solved by multi-objective optimization. Equation (31) is the multi-objective function. $w(\text{DSO})$ and $w(\text{DOPF})$ are two weights of two objective functions.

$$\min_{x_a(t) \& P_g(t)} w(\text{DSO}) \sum_{t=0}^{T} \sum_{a} c_a[x_a(t)] + w(\text{DOPF}) \sum_{t=0}^{T} \sum_{g=1}^{G} c_g[P_g(t)] \tag{31}$$

## 3. Results

### 3.1. Parameter Settings

Figure 6 gives the TN of the case. There are 4 nodes, 10 links, and 2 FCSs in the TN. A total of 1 h is taken as a time interval. The travel cost $c_a[x_a(t)]$ is the time cost. Table 3 provides the parameters of this TN case.

**Table 3.** Distance of links.

| | *a* | *a′* | *b* | *b′* | *c* | *c′* | *d* | *d′* | *e* | *e′* |
|---|---|---|---|---|---|---|---|---|---|---|
| $d_a$ (km) | 120 | 120 | 120 | 120 | 120 | 120 | 120 | 120 | 240 | 240 |
| $d_a^{\gamma}$ (km) | 0 | 120 | | | | | | | 120 | 120 |

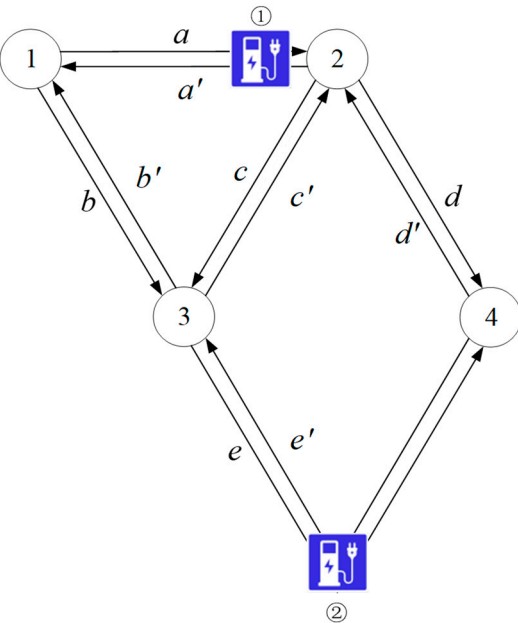

**Figure 6.** Case of TN.

Table 3 describes the distance of the links. The distances $d_a$ of link $e$ and $e'$ are 240 km. The other links are all 120 km. This means that vehicles passing $e$ and $e'$ at the speed of 120 km/h need 2 h. Passing the other 8 links needs 1 h. FCS ① is located at the exit of link $a$ and the entrance of link $a'$. The distance from ① to the exit of link $a$ is 0 and to the exit of link $a'$ is 120 km. FCS ② is located at the middle of link $e$ and $e'$. The distance from ② to the exits of link $e$ and $e'$ is 120 km each.

Table 4 provides the attributes of the vehicle types. One type of GV and one type of EV are set. The battery capacity of the EV is 60 kWh and the highest available speed in the TN is 120 km/h.

**Table 4.** Attributes of the vehicle types.

| $m$ | Type | $E_m$ (kWh) | $sp_m$ (km/h) |
|---|---|---|---|
| 1 | GV | 0 | 120 |
| 2 | EV | 60 | 120 |

Figure 7 provides the PDN of the case. This case is adopted from [42] and the 2 constant loads in bus 2 and bus 7 are substituted with 2 FCSs. There is not a real or reactive ramp constraint on the power source in bus 51. This is a radial PDN. Its network structure is in accordance with most common PDNs. In the same power transmission direction, a greater power loss would be caused if the load was farther from the bus because the power source would need to flow over more branches in order to reach the load and, therefore, each branch would experience some power loss.

Table 5 provides the connection of the FCS. FCS ① is down links $a$ and $a'$ which are shorter than links $e$ and $e'$ where FCS ② is. However, FCS ② is closer to the power source (bus 51) and FCS ① remoter. The charging power of the two FCSs is 30 kW each and the maximum vehicle number for one side $x^{ch}_{amax}$ is 15 (there are two sides to one FCS = 30 vehicles). For vehicle 2 (EV), the single charging time is 2 h (2 intervals).

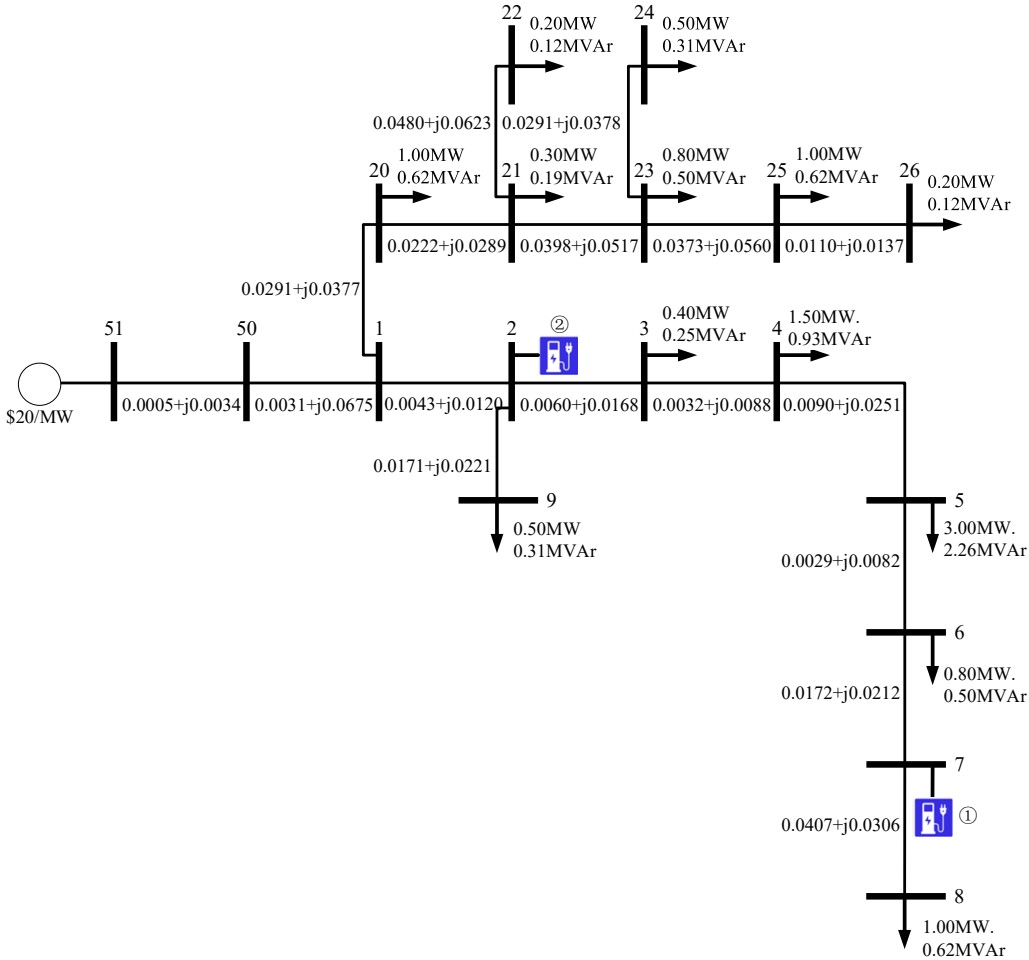

**Figure 7.** Case of the PDN.

**Table 5.** Parameters of FCSs.

| Station | Bus | $p_i^{ch}$ (kW) | Link 1 | $x_{1max}^{ch}$ | $d_1^\gamma$ (km) | Link 2 | $x_{2max}^{ch}$ | $d_2^\gamma$ (km) |
|---|---|---|---|---|---|---|---|---|
| ① | 7 | 30 | $a$ | 15 | 0 | $a'$ | 15 | 120 |
| ② | 2 | 30 | $e$ | 15 | 120 | $e'$ | 15 | 120 |

If vehicles are assigned to ①, the total travel cost of the DSO is less but the total cost of power generation would be higher in the same power flow path (in this case, it is buses 51–50–1–2–3–4–5–6–7–8); the bus remoter to the power source causes higher power loss and power flowing to more branches. On the contrary, if vehicles are assigned to ②, the total cost of power generation would be less but the total travel cost would be higher.

Table 6 provides the link-path incidence table. A total of 2 O–D pairs, 1-4 and 4-1, are set. Each O–D pair includes two tasks. One is assigned in time period 0 to 10. The other is assigned in time period 4 to 14. Each task includes 40 GVs ($m = 1$) and 20 EVs ($m = 2$). Each vehicle type has four paths. The part on the right side of the column path $q$ is the link-path incidence table. The empty cells in the links columns from column $a$ to $e'$ are logical 0 representing the path that does not pass through the link. Number 1 and 2 in the links columns from column $a$ to $e'$ are logical 1, which are the paths passing through the link. While a cell is at 1, it merely passes through it. While it is at 2, it passes through the link and charges at the FCS on the link.

The weight of the case keeps $w(DSO) + w(DOPF) = 1$.

Finally, the case is operated by MATLAB with GUROBI QCP multi-objective optimization solver. The following are the results and analysis.

**Table 6.** Link-path incidence table.

| rs | h(t) | m ($F^{rs}_{hm}$) | q | a | a' | b | b' | c | c' | d | d' | e | e' |
|----|------|------|---|---|----|---|----|---|----|---|----|---|----|
| 14 | 1 (0–10) | 1(40) | 1 | 1 |  |  |  |  |  | 1 |  |  |  |
|  |  |  | 2 | 1 |  |  |  | 1 |  |  |  | 1 |  |
|  |  |  | 3 |  |  | 1 |  |  | 1 | 1 |  |  |  |
|  |  |  | 4 |  |  | 1 |  |  |  |  |  | 1 |  |
|  |  | 2(20) | 1 |  | 2 |  |  |  |  | 1 |  |  |  |
|  |  |  | 2 |  | 2 |  |  | 1 |  |  |  | 1 |  |
|  |  |  | 3 | 1 |  |  |  | 1 |  |  |  | 2 |  |
|  |  |  | 4 |  |  |  | 1 |  |  |  |  | 2 |  |
|  | 2 (4–14) | 1(40) | 1 | 1 |  |  |  |  |  | 1 |  |  |  |
|  |  |  | 2 | 1 |  |  |  | 1 |  |  |  | 1 |  |
|  |  |  | 3 |  |  | 1 |  |  | 1 | 1 |  |  |  |
|  |  |  | 4 |  |  | 1 |  |  |  |  |  | 1 |  |
|  |  | 2(20) | 1 |  | 2 |  |  |  |  | 1 |  |  |  |
|  |  |  | 2 |  | 2 |  |  | 1 |  |  |  | 1 |  |
|  |  |  | 3 | 1 |  |  |  | 1 |  |  |  | 2 |  |
|  |  |  | 4 |  |  |  | 1 |  |  |  |  | 2 |  |
| 41 | 1 (0–10) | 1(40) | 1 |  | 1 |  |  |  |  |  | 1 |  |  |
|  |  |  | 2 |  |  |  | 1 |  | 1 |  | 1 |  |  |
|  |  |  | 3 |  | 1 |  |  |  | 1 |  |  |  | 1 |
|  |  |  | 4 |  |  |  | 1 |  |  |  |  |  | 1 |
|  |  | 2(20) | 1 |  | 2 |  |  |  |  |  | 1 |  |  |
|  |  |  | 2 |  | 1 |  |  |  | 1 |  |  |  | 2 |
|  |  |  | 3 |  | 2 |  |  |  | 1 |  |  |  | 1 |
|  |  |  | 4 |  |  |  | 1 |  |  |  |  |  | 2 |
|  | 2 (4–14) | 1(40) | 1 |  | 1 |  |  |  |  |  | 1 |  |  |
|  |  |  | 2 |  |  |  | 1 |  | 1 |  | 1 |  |  |
|  |  |  | 3 |  | 1 |  |  |  | 1 |  |  |  | 1 |
|  |  |  | 4 |  |  |  | 1 |  |  |  |  |  | 1 |
|  |  | 2(20) | 1 |  | 2 |  |  |  |  |  | 1 |  |  |
|  |  |  | 2 |  | 1 |  |  |  | 1 |  |  |  | 2 |
|  |  |  | 3 |  | 2 |  |  |  | 1 |  |  |  | 1 |
|  |  |  | 4 |  |  |  | 1 |  |  |  |  |  | 2 |

Empty cells are logical 0 representing the path that does not pass through the link. Paths 1 and 2 are logical 1, which are the paths passing through the link. While a cell is at 1, it merely passes through it. While a cell is at 2, it passes through the link and charges at the FCS on the link, which is $\sum_{q \in i}$ in Equation (28).

### 3.2. Results and Analysis

3.2.1. Optimal Solution versus Weight

Figure 8 provides the objective value of the DSO and DOPF versus $w$(DSO) for time period 0 to 14. One is the left parallel region where $w$(DSO) $\leq 0.035$ and PDN are prior. The second is the middle compromise region where $0.035 < w$(DSO) $\leq 0.041$. The third is the right parallel region where $w$(DSO) $> 0.041$ and the TN is prior.

3.2.2. Optimal Solution While TN Prior

Figure 9 provides the instantaneous flow of 2 tasks for time period 0 to 10 and 4 to 14, respectively, while TN is prior. The optimal result of the objective function is 640 h and $3573.41. The positive bars represent instantaneous departing flow. The negative bars are instantaneous arriving flow. The two tall bars represent instantaneous GVs. The assignment of GVs is random in some time intervals because there are no specific constraints about GV. In both time periods, 0 to 10 and 4 to 14, the assignment of EVs separates in most time intervals. It does not accumulate to a few specific time intervals like GVs. This assignment pattern decreases the power loss.

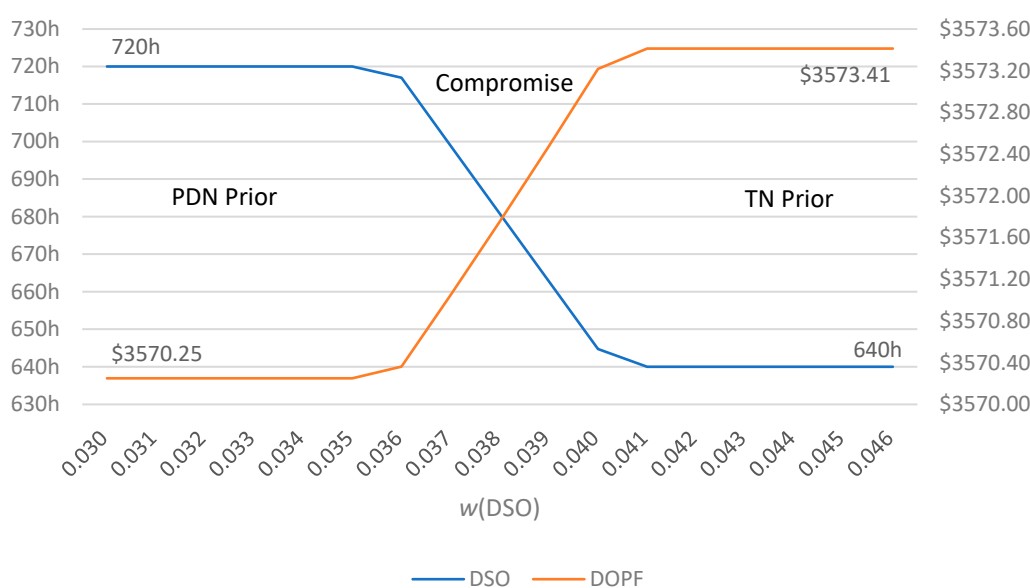

**Figure 8.** Optimal solution while T = 14. h represents time unit hours.

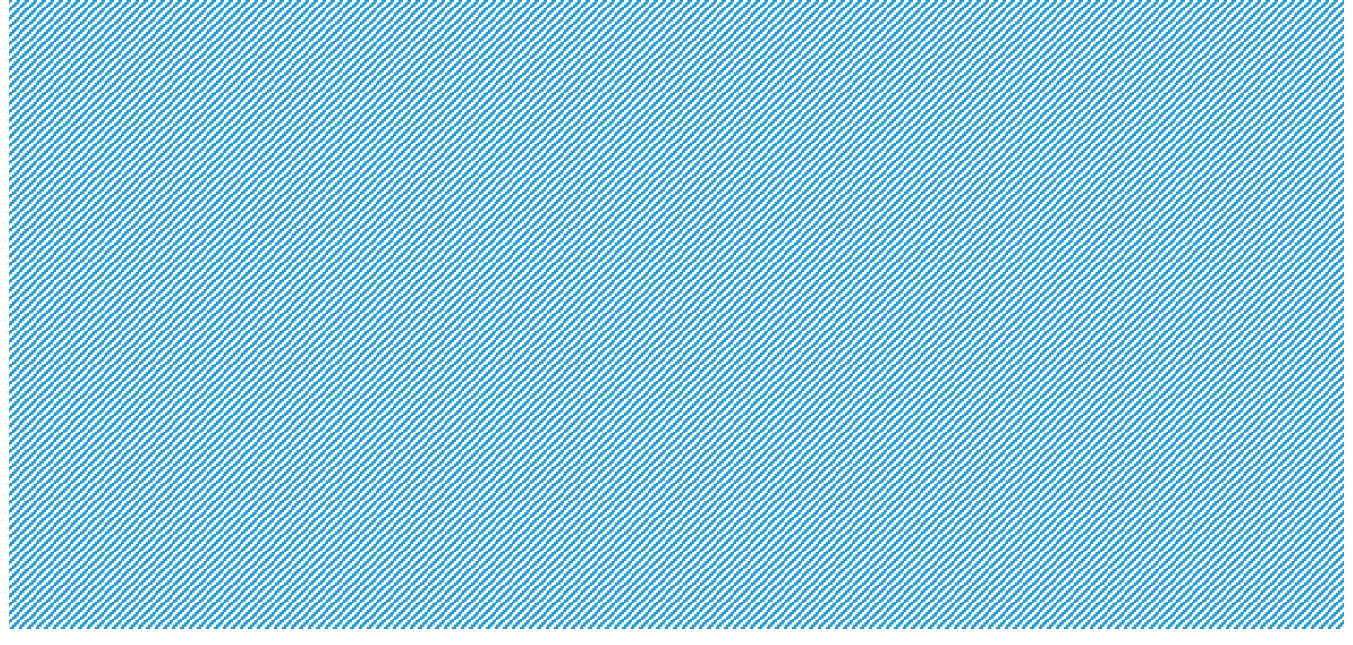

**Figure 9.** (**a**) Instantaneous flow rate of task 1 for time period 0 to 10 under TN prior; (**b**) instantaneous flow rate of task 2 for time period 4 to 14 under TN prior. (G represents GV, E represents EV, 14 and 41 are O–D pairs, r represents departing flow, and s represents arriving flow).

Figure 10 provides the path assignment. The blue bars represent the instantaneous departing flow assigned to paths passing through station ①. The blue line represents the real power demand of station ①. The orange bars represent the instantaneous departing flow assigned to paths passing through station ②. The orange line represents the real power demand of station ②. All EVs are assigned to path 1①24 and 42①1. All EVs pass through station ①.

### 3.2.3. Optimal Solution While PDN Prior

While PDN is prior, the optimal results of the objective function are 720 h and $3570.25.

Figure 11 gives the instantaneous flow while PDN is prior. EVs are still separated by various time intervals in both tasks.

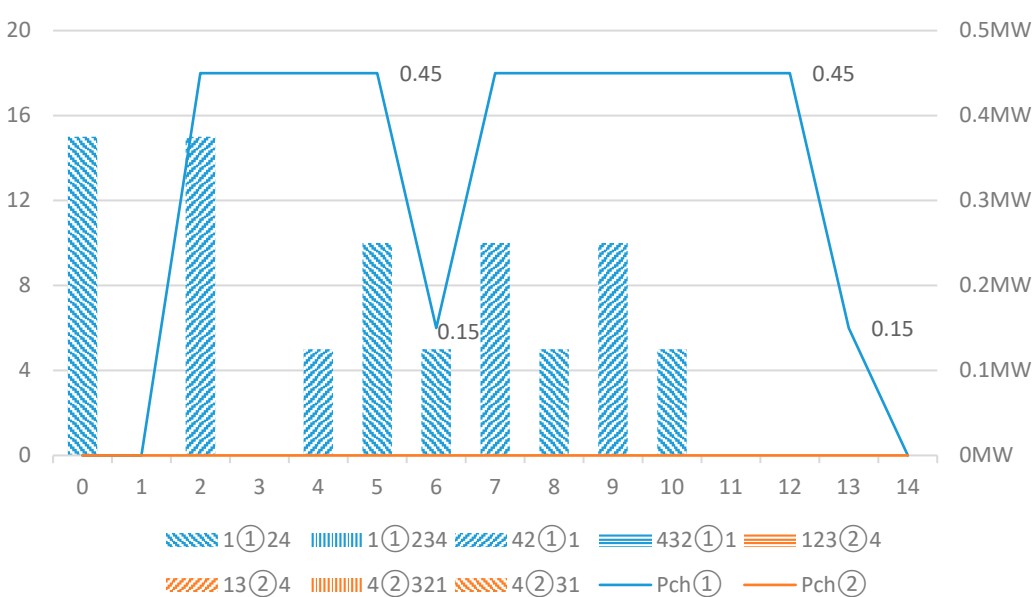

**Figure 10.** Path assignment of EVs of 2 tasks *h* (TN prior, T = 14).

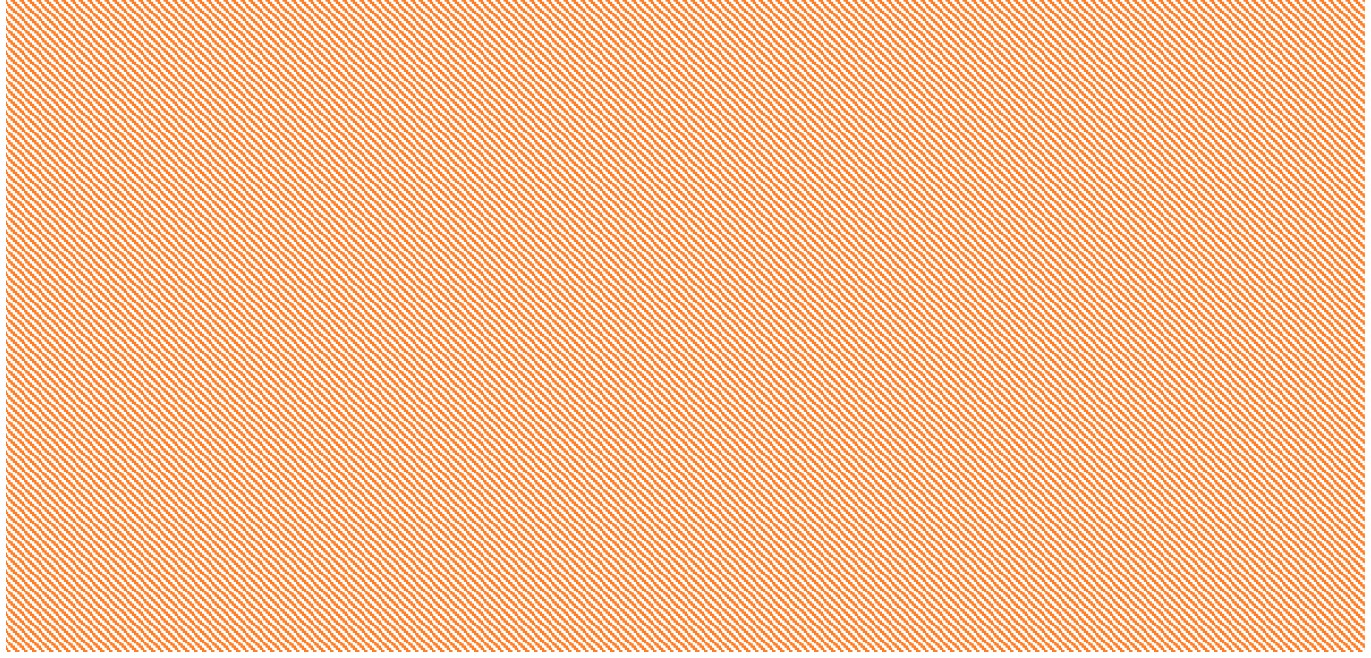

**Figure 11.** (**a**) Instantaneous flow of task 1 for time period 0 to 10 under PDN prior; (**b**) instantaneous flow of task 2 for time period 4 to 14 under PDN prior.

Figure 12 provides the path assignment while PDN is prior. For TN prior, the result of the path assignment is the opposite. All EVs are assigned to pass through station ②.

### 3.2.4. Changing Charging Power of FCS ②

Finally, while the charging power of station ② on link *e* and *e'* increases to 60 kW/h, the EVs with 60 kWh charging capacity require only one hour for charging at station ②. Then, the travel times of path 1①24, 23②4, 42①1, and 4②32 are equal to 4 h. The optimal results are constant at 640 h and $3570.25, not following the change in weight.

The last result can be considered while electricity operators update charging piles. It is not possible to ensure optimal matching of link and bus connections unless there is a precise plan while the transportation network and the PDN are being updated and constructed in

the meantime. Most links are constructed earlier than FCSs. The order of updating affects the optimal result of the cooperation.

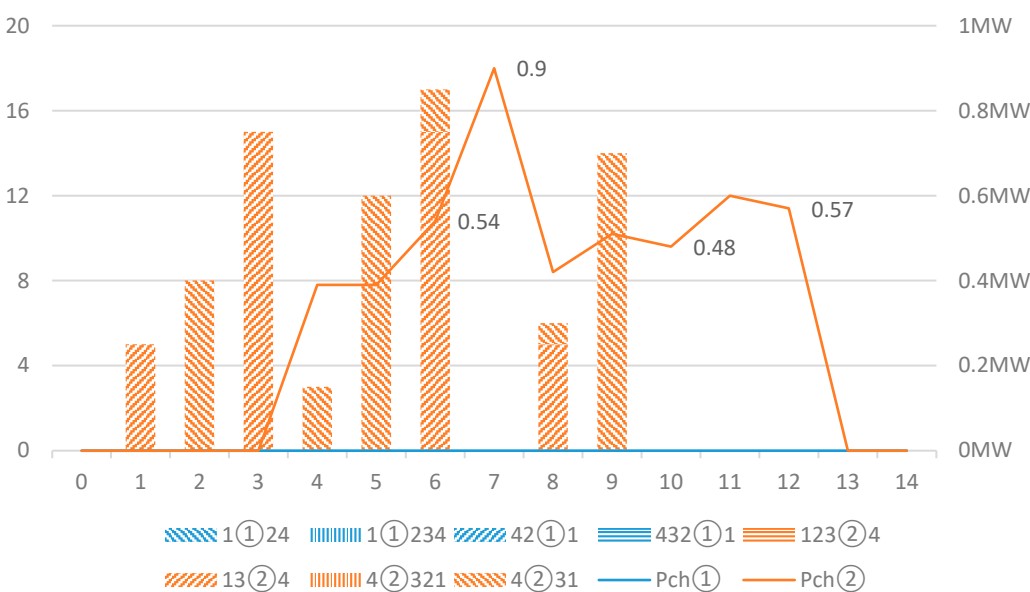

**Figure 12.** Path assignment of EVs of 2 tasks *h* (PDN prior, T = 14).

## 4. Conclusions

This project studied the cooperation between dynamic TNs and dynamic PDNs. The dynamic TN model used adopted a DSO model based on the Wardrop II principle. The dynamic PDN model used adopted the DOPF model. EVs and FCS are designed to connect to both networks. We used blended multi-objective optimization to realize the cooperation. The DSO model guarantees SFIFO constraints and considers multiple O–D pairs, multiple tasks, and multiple vehicle types. Path *q* is the minimum assignment class. Vehicles are assigned to various time intervals and paths. The DOPF contains linear and quadratic constraints, relaxing the voltage phase angle. To most power systems ignoring branch capacity, this is a useful and simpler optimization solver, which decreases the difficulty of operation. The FCS model considers the accumulation of vehicles and the influence of power demand. In general, it has two main novelties compared with former literature studies. One is to provide a solution to the lack of DSO in the TN and PDN dynamic cooperation research area. The other is that this DSO model is considered multi-task, which is not the case with common DSO models.

The result includes three regions: PDN prior region, PDN and TN compromise region, and TN prior region. The optimal solutions for the objective function change with the change in the weight of the two objective functions. The result displays that while TN is prior, EVs are assigned to paths with lower travel cost as much as possible. While PDN is prior, EVs are assigned to paths directed through FCSs near to the bus of the power source as much as possible to decreases the power loss due to the length of the transmission line. In the former two scenarios, the assignment of EVs cover most time intervals. This dispersive assignment strategy can also decrease power loss in time intervals. The charging power can be changed, which could provide some beneficial information to electricity operators while updating charging piles. This project can be used in logistics transportation under traffic restriction.

In future, this work can be converted into a dynamic cooperation model with DUO and DOPF by adjusting the DSO model. It will need to consider the congestion caused by users' spontaneous behaviors in FPC. An appropriate DUO objective function should be adopted to describe different scenarios of traffic equilibrium. Voltage phase angle can also

be taken into account in the DOPF model if a more complex optimization solver is used to operate this model.

**Author Contributions:** Methodology, Z.C.; Resources, F.X., S.L. and L.J.; Writing—original draft, Z.C.; Writing—review and editing, Z.C.; Supervision, B.H. Project administration, B.H. and S.L.; Funding acquisition, B.H. and S.L. All authors have read and agreed to the published version of the manuscript.

**Funding:** This research is supported and sponsored in part by the Research Development Fund RDF-21-01-061 at Xi'an Jiaotong-Liverpool University and the 2022 Fundamental and Applied Fundamental Research Project of Guangdong Province Basic Research Programme: Research on demand side scheduling and optimization in electrified city transportation systems (the project code is currently not available).

**Data Availability Statement:** Not applicable.

**Conflicts of Interest:** The authors declare no conflict of interest.

## Nomenclature and Abbreviations

*Notations in the Transportation Network Model*

Sets and Indices

| | |
|---|---|
| $a$ | Link |
| $A_l$ | Set of links whose tail node is $l$ ($l \neq r \cup l \neq s$) |
| $A_r$ | Set of links whose tail node is origin node $r$ |
| $A_s$ | Set of links whose tail node is destination node $s$ |
| $B_l$ | Set of links whose head node is node $l$ ($l \neq r \cup l \neq s$) |
| $B_r$ | Set of links whose head node is origin node $r$ |
| $B_s$ | Set of links whose head node is destination node $s$ |
| $c_a[x_a(t)]$ | Travel cost on link $a$ |
| $h$ | Task |
| $l$ | Node excluding origin $r$ and destination $s$ |
| $m$ | Vehicle type |
| $q$ | Path |
| $r$ | Origin node |
| $s$ | Destination node |
| $t$ | Time interval index |
| $t_{am}^{ch}$ | Charging time interval index of vehicle $m$ at the charging station on link $a$ |

Parameters

| | |
|---|---|
| $d_a$ | Distance of link $a$ |
| $E_m$ | Battery capacity of electric vehicle $m$ |
| $F_{mh}^{rs}$ | Total number of vehicle $m$ in task $h$ with origin $r$ and destination $s$ |
| $F_{rmh}^{rs}$ | Total number of vehicle $m$ in task $h$ departing from origin $r$ towards destination $s$ |
| $F_{smh}^{rs}$ | Total number of vehicle $m$ in task $h$ arriving at destination $s$ from origin $r$ |
| $sp_m$ | Speed of vehicle $m$ |
| $\tau_{am}$ | Link $a$ travel time of vehicle $m$ without congestion |
| T | Final time |
| $w$(DSO) | Weight of dynamic system optimal objective function |

*Variables*

| | |
|---|---|
| $f_{rmh}^{rs}(t)$ | Instantaneous departing flows of vehicle $m$ number in task $h$ departing from origin $r$ towards destination $s$ at time $t$ |
| $f_{smh}^{rs}(t)$ | Instantaneous arriving flows of vehicle $m$ number in task $h$ arriving at destination $s$ from origin $r$ at time $t$ |
| $u_a(t)$ | Total inflow rate on link $a$ over path $q$ at time $t$ |
| $u_{amhq}^{rs}(t)$ | Inflow rate on link $a$ over path $q$ which belongs to vehicle $m$ number in task $h$ from origin $r$ and destination $s$ at time $t$ |
| $x_a(t)$ | Total number of vehicles travelling on link $a$ at time $t$ |
| $x_{amhq}^{rs}(t)$ | Number of vehicles on link $a$ over path $q$ which belong to vehicle $m$ number in task $h$ with origin $r$ and destination $s$ at time $t$ |

| | |
|---|---|
| $v_a(t)$ | Total exit flow rate from link $a$ at time $t$ |
| $v_{amhq}^{rs}(t)$ | Exit flow rate from link $a$ over path $q$ which belongs to vehicle $m$ number in task $h$ with origin $r$ and destination $s$ at time $t$ |

*Notations in the Power Distribution Network Model*

Sets and Indices

| | |
|---|---|
| $fb$ | From bus |
| $g$ | Generating unit |
| $i$ | Bus i |
| $j$ | Bus $j$ |
| $k$ | Bus $k$ |
| $tb$ | To bus |

Parameter

| | |
|---|---|
| $b_{ij}$ | Susceptance at branch from bus $i$ to bus $j$ |
| $b_j$ | Sum of susceptance caused by shunt capacitors at bus $j$ |
| $fb$ | From bus |
| $P_{gmin}$ | Minimum real power output of unit $g$ |
| $P_{gmax}$ | Maximum real power output of unit $g$ |
| $P_j^d$ | Real load at bus $j$ |
| $P_{ramp}$ | Ramp of real power output per time frame |
| $Q_{gmin}$ | Minimum reactive power output of unit $g$ |
| $Q_{gmax}$ | Maximum reactive power output of unit $g$ |
| $Q_j^d$ | Reactive load at bus $j$ |
| $Q_{ramp}$ | Ramp of reactive power output per time interval |
| $r_{ij}$ | Resistance of branch from bus $i$ to bus $j$ |
| $S_{ijmax}$ | Complex power flow limit of branch $ij$. |
| $T_{am}^{ch}$ | Charging time of vehicle $m$ in charging station on link $a$ |
| tb | To bus |
| $V_{imin}^2$ | Minimum value of square of voltage magnitude at bus $i$ |
| $V_{imax}^2$ | Maximum value of square of voltage magnitude at bus $i$ |
| $w$(DOPF) | Weight of dynamic optimal power flow objective function |
| $x_{ij}$ | Reactance of branch from bus $i$ to bus $j$ |
| $z_{ij}$ | Impedance of branch from bus $i$ to bus $j$ |

Variables

| | |
|---|---|
| $c_g[P_g(t)]$ | Cost of real power output of unit $g$ at time $t$ |
| $\dot{I}_{ij}(t)$ | Complex current in branch from bus $i$ to bus $j$ at time $t$ |
| $\dot{I}_{ij}^*(t)$ | Conjugate complex current in branch from bus $i$ to bus $j$ at time $t$ |
| $I_{ij}^2(t)$ | Square of current magnitude in branch from bus $i$ to bus $j$ at time $t$ |
| $P_g(t)$ | Real power output of unit $g$ at time $t$ |
| $P_{ij}(t)$ | Real power flow of branch from bus $i$ to bus $j$ at time $t$ |
| $P_{jk}(t)$ | Real power flow of branch from bus j to bus $k$ at time $t$ |
| $P_j^g(t)$ | Real power output of unit g at bus $j$ at time $t$ |
| $Q_g(t)$ | Reactive power output of unit $g$ at time $t$ |
| $Q_{ij}(t)$ | Reactive power flow of branch from bus $i$ to bus $j$ at time $t$ |
| $Q_{jk}(t)$ | Reactive power flow of branch from bus $j$ to bus $k$ at time $t$ |
| $Q_j^g(t)$ | Reactive power generation of unit $g$ at bus $j$ at time $t$ |
| $\widetilde{S}_{ij}(t)$ | Complex power flow of branch from bus $i$ to bus $j$ at time $t$ |
| $\dot{V}_i(t)$ | Complex voltage at bus $i$ at time $t$ |
| $\dot{V}_j(t)$ | Complex voltage at bus $j$ at time $t$ |
| $V_i^2(t)$ | Square of voltage magnitude at bus $i$ at time $t$ |
| $V_j^2(t)$ | Square of voltage magnitude at bus $j$ at time $t$ |

*Notations in the Fast Charging Station Model*

Parameter

| | |
|---|---|
| $d_a^\gamma$ | Distance from the charging station on link $a$ to the link's exit |
| $p_i^{ch}$ | Real power of charging pile in charging station on link $a$ |
| $\tau_{am}^\gamma$ | Link travel time of vehicle $m$ from the charging station on link $a$ to the link's exit without congestion |

| | |
|---|---|
| $x_{amax}^{ch}$ | Maximum number of vehicles charging in charging station on link $a$ |
| Variables | |
| $P_i^{ch}(t)$ | Real power demand of charging station connecting to bus $i$ at time $t$ |
| $Q_j^{ch}(t)$ | Reactive power demand of charging station connecting to bus $j$ at time $t$ |
| $x_a^{ch}(t)$ | Number of vehicles staying in charging station on link $a$ |
| *Abbreviation* | |
| AC | Alternating current |
| DOPF | Dynamic optimal power flow |
| DSO | Dynamic system optimal |
| DTA | Dynamic traffic assignment |
| DUO | Dynamic user optimal |
| EV | Electric vehicle |
| FCC | Flow conservation constraints |
| FCS | Fast charging station |
| FPC | Flow propagation constraints |
| GV | Gasoline vehicle |
| LFC | Load frequency control |
| LP | Linear programming |
| MILP | Mix-integer linear programming |
| MIQCP | Mix-integer quadratically constrained programming |
| O-D | Origin–destination |
| OPF | Optimal power flow |
| PDN | Power distribution network |
| PEV | Plug-in electric vehicle |
| PSO | Particle swarm optimization |
| SFIFO | Strong first-in-first-out |
| QCP | Quadratically constrained programming |
| SO | User optimal |
| SOC | State of charge |
| TA | Traffic assignment |
| TAP | Traffic assignment problem |
| TN | Transportation network |
| UE | User equilibrium |

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
