# Peer review of "Dynamic Cooperation of Transportation and Power Distribution Networks via EV Fast Charging Stations"

_wevj, doi:10.3390/wevj14020038_

Round 1
Reviewer 1 Report
I have some comments that have to be considered in the modified manuscript. Please find attached PDF file.

Reviewer 2 Report
1. Figure 1 is too big.
2. The review on charging methods and especially the “fast charging” is far from enough. Please investigate more relevant state-of-the-art research, intrusion-detector-dependent distributed economic model predictive control for load frequency regulation with PEVs under cyber attacks, resilient distributed frequency estimation for PEVs coordinating in load frequency regulation under cyber attacks.
3. Figure 2 is not properly displayed.
4. Please make sure all symbols in Equations are explained. In other words, give their physical meanings below each equation.
5. Figure 3 is confused. Please give more explanations. The similar problems also exist in Figure 4.
6. I suggest adding a framework to incorporate the whole content of the paper, maybe at the very beginning of the work, to make readers more clear about the relationships of each part.
7. Grammar mistakes exist, for example, “Equation (15) is an constraint”. Please check the whole paper.
8. As a review paper, future works must be included. Please add the promising future works at the end of the paper.
Reviewer 3 Report
The paper is generally well-written and supported by a substantial amount of literature sources. I have the following comments:
1. It needs to be clarified if the paper is a review or research paper.
2. Please clarify the aim and the novelty at the end of part 1.
3. Please clarify the origin of the mathematical model presented in part 2. You may want to add citations.
4. Fig. 8 can be supported with better explanations.
5. You may want to demonstrate the applied multi-objective optimisation with a specific case study.
Thank you for the interesting paper.
